# Genome-Wide Characterization and Expression Analysis of Fatty acid Desaturase Gene Family in Poplar

**DOI:** 10.3390/ijms231911109

**Published:** 2022-09-21

**Authors:** Hui Wei, Ali Movahedi, Songzhi Xu, Yanyan Zhang, Guoyuan Liu, Soheila Aghaei-Dargiri, Mostafa Ghaderi Zefrehei, Sheng Zhu, Chunmei Yu, Yanhong Chen, Fei Zhong, Jian Zhang

**Affiliations:** 1Key Laboratory of Landscape Plant Genetics and Breeding, School of Life Sciences, Nantong University, Nantong 226001, China; 2College of Biology and the Environment, Nanjing Forestry University, Nanjing 210037, China; 3College of Arts and Sciences, Arlington International University, Wilmington, DE 19804, USA; 4Department of Horticulture, Faculty of Agriculture and Natural Resources, University of Hormozgan, Bandar Abbas 7916193145, Iran; 5Department of Animal Science, Faculty of Agriculture, Yasouj University, Yasouj 7591874831, Iran

**Keywords:** FAD, UFA, poplar, abiotic stress

## Abstract

Fatty acid desaturases (FADs) modulate carbon–carbon single bonds to form carbon–carbon double bonds in acyl chains, leading to unsaturated fatty acids (UFAs) that have vital roles in plant growth and development and their response to environmental stresses. In this study, a total of 23 *Populus trichocarpa*
*FAD (PtFAD)* candidates were identified from the poplar genome and clustered into seven clades, including FAB2, FAD2, FAD3/7/8, FAD5, FAD6, DSD, and SLD. The exon–intron compositions and conserved motifs of the PtFADs, clustered into the same clade, were considerably conserved. It was found that segmental duplication events are predominantly attributable to the *PtFAD* gene family expansion. Several hormone- and stress-responsive elements in the *PtFAD* promoters implied that the expression of the *PtFAD* members was complicatedly regulated. A gene expression pattern analysis revealed that some *PtFAD* mRNA levels were significantly induced by abiotic stress. An interaction proteins and gene ontology (GO) analysis indicated that the PtFADs are closely associated with the UFAs biosynthesis. In addition, the UFA contents in poplars were significantly changed under drought and salt stresses, especially the ratio of linoleic and linolenic acids. The integration of the PtFAD expression patterns and UFA contents showed that the abiotic stress-induced PtFAD3/7/8 members mediating the conversion of linoleic and linolenic acids play vital roles in response to osmotic stress. This study highlights the profiles and functions of the PtFADs and identifies some valuable genes for forest improvements.

## 1. Introduction

Fatty acids (FAs), the hydrophobic interior of cell components, are ubiquitous in plants and participate in many physiological processes [1]. The form of FA in a cell is mainly bound and rarely exists in a free state. Their biosynthesis and regulation play an important role in the basic metabolic activities of cells [2]. The FAs can be divided into saturated FAs (SFAs) and unsaturated FAs (UFAs), according to the presence or absence of a double bond of FAs, and the UFAs are classified into the monounsaturated FAs (MUFAs), with one double bond, and the polyunsaturated FAs (PUFAs), containing two or more double bonds [3]. Also, based on the position of the unsaturated bond, the UFAs can be divided into a series of ω-3, ω-6, ω-7, and ω-9 UFAs [4]. The UFAs are essential for bio-membranes and improve membrane fluidity [5,6]. FA desaturases (FADs) can catalyze single-bond UFAs into double-bond UFAs in fatty acyl chains [7]. Two categories of FADs have been identified in plants; one is the soluble FAD, broadly distributed on plant cell plastids, and the other is the membrane-bound FAD, mainly localized on the endoplasmic reticulum (ER) plastid membranes [8]. It has been reported that the soluble FAD can be identified as stearoyl-ACP desaturase (FAB2 or SAD), which explicitly converts stearoyl-ACP (C_18:0_) into palmitoleic acid and oleic acid (C_18:1_). Then, the Δ12 FADs control the oleic acid to linoleic acid, and the Δ15 FADs or Δ6 FADs further convert the linoleic acid into α-linolenic acid or γ-linolenic acid, respectively [9].

There are two spatially independent pathways for glycerolipid biosynthesis in advanced plants, including the prokaryotic and eukaryotic pathways. The prokaryotic pathway is localized in the chloroplast, while the eukaryotic pathway refers to the glyceride biosynthesis in the ER [10]. The glycolytic pathway provides acetyl-coenzyme A (CoA) as the FA precursor [11]. Acetyl-CoA carboxylase (ACC) converts two molecules of the acetyl-CoA into malonyl-CoA, and then the malonyl-CoA can be catalyzed to form the acyl carrier protein (ACP). The C_18:0_-ACP as the substrate is catalyzed to generate phosphatidylglycerol (PG), and the PG can be converted to diacylglycerol (DG) with the catalyzation phosphatase. For example, the FAD5-8 modulate the PG, sulfoquinovosyl diacylglycerol (SQDG), and digalactosyldiacylglycerol (DGDG) to form C_18:3_ linolenate [12]. In the eukaryotic pathway, the C_16:0_-ACP and C_18:1_-ACP can be catalyzed by the acyl-CoA synthase to form C_16:0_-CoA and C_18:1_-CoA, and they are transferred by the acyl-CoA binding protein (ACBP) to the ER for the phosphatidylic acid (PA), phosphatidylcholine (PC), and another glyceride biosynthesis [13]. Therefore, the C_18_ UFAs are synthesized by the prokaryotic and eukaryotic pathways, while the C_16_ UFAs can be generated by the eukaryotic pathway [10].

The studies on desaturases in large numbers of plant species have revealed their characterizations and functions. For example, the glycerolipid n-3 desaturase has been revealed to be responsible for lipid composition and has a significant function in the biosynthesis of the trienoic FAs (TFAs), including the C_16:3_ and C_18:3_ FAs [14]. The Arabidopsis omega-3 (ω-3) FAD has been demonstrated to be limited to linolenic acid (C_18:3_) production in seeds [15]. A mutation in the *FAD7* in Arabidopsis reduced the TFA contents, and a mutation at the *fad8* locus of the Arabidopsis showed lower levels of phospha-tidylglycerol and sulfoquinovosyldiacylglycerol under a low temperature [16]. The Arabidopsis FAD7 and FAD8 were identified as functionally related, and the constitutive expression of the *FAD8* in the mutation of the *FAD7* plants resulted in a genetic complementation of the mutation [17]. Shah et al. [18] proved that up-regulation of the *FAD3* transcript levels leads to increasing levels of the C_18:3_ FAs and correspondingly decreasing levels of the C_18:2_ FAs. The FAD2, a plant oleate desaturase, is responsible for PUFA biosynthesis, especially linoleate and linolenate [19]. The FAD2 and FAD3 are exposed on the cytosolic side of the ER membranes and catalyze the conversion of the C_18:1_ PC-bound oleate acid to the C_18:2_ linoleic acid and C_18:3_ linolenic acid, respectively [20]. In addition, UFAs have been identified to play an essential role in responding to abiotic stresses, such as low and high temperature, salt, drought, and heavy metals [21,22]. An increased UFA content in the chloroplast membrane enhanced the Arabidopsis tolerance to chilling, freezing, and oxidative stress [23]. The *Oryza sativa* FAD2 (OsFAD2) participated in the FA desaturation and promoted the tolerance of rice to low-temperature stress [24]. The *Brassica napus* FAD3 (BnFAD3) and *Arabidopsis thaliana* FAD8 (AtFAD8) were proved to promote the linolenic acid/linoleic acid ratio and resistance to osmotic stress [25]. The ratio of linolenic acid to linoleic acid was significantly increased when the cowpea was treated with drought stress [26]. Moreover, FADs are crucial for regulating UFAs, which may be associated with limiting the compositions and contents of the UFAs, especially linoleic and linolenic acids. Previous studies have indicated that *FAD* genes are essential for plant growth and development and the sustenance of plants resilience to different environmental stresses. The FAD3, FAD7, and FAD8 had close associations with TFAs, the precursor of oxylipin, which modulated Arabidopsis pollen development [27]. The post-translational stability of Arabidopsis FAD8 was considered an essential regulatory mechanism in response to temperature, modulating the TFA contents at high temperatures [28]. The overall mRNA levels of *Zea mays FAD7/8* (*ZmFAD7* and *ZmFAD8*) were elevated upon exposure to low temperature, and *the ZmFAD7*/*8* expression was accumulated in the roots when the maize was treated with high-salt stress [29]. A low temperature induced the soybean *GmFAD3A* gene, and overexpression of it in rice promoted the cold resistance and seed germination rate [30,31]. The *AtFAD3* or *AtFAD8* was ectopically overexpressed in tobacco, increasing the osmotic stress resistance [9]. The tomato FAD3 has been reported to have crucial roles in resistance to salt stress, while down-regulation of the tomato *FAD7* expression dramatically improved its tolerance under high-temperature stress [32]. The Arabidopsis FAD2 and FAD6 were identified to have a positive function in response to salinity stress [33,34]. Also, the mutant of *ads2* enhanced sensitivity to chilling and freezing treatments [35], and the mutant of *sld* showed an increased sensitivity to low-temperature stress [36]. In tobacco, the expression levels of *FAD3* and *8* were promoted under drought stress, and the *FAD7* overexpression significantly improved the drought tolerance [25]. Thus, the *FADs,* as essential regulatory genes, can respond to various stresses and are associated with alleviating the damage induced by various stresses.

Plant genomes contributed to the genome-wide analysis of the *FAD* gene family. In Arabidopsis and rice, 24 and 18 *FAD* genes were identified, including soluble and membrane-bound FADs [8]. Also, 68 putative FAD members were identified in *Brassica napus;* 23 *FAD* genes were retrieved from the *Cucumis sativus* genome; and 19 *FAD* members were found in *Gossypium raimondii* [37,38,39]. In addition, 20 full-length *FAD* members were identified from *Medicago truncatula*, and 30 putative *FADs* were found in the maize genome, including 17 membrane-bound and 13 soluble *FADs* [13,40]. For *Triticum aestivum*, 68 *FAD* genes were retrieved in the wheat genome, and 29 *FAD* genes were found in the *Cucumis sativus* genome [41,42]. Poplar, a major source of natural material for industrial production, is widely cultivated worldwide. Here, we identified the genome-wide identification of the *Populus trichocarpa FAD* members and divided them into two subfamilies, or seven clades, based on the phylogenetic tree. Also, we investigated the chromosome distribution, collinearity relationship, selection pressure, gene structure, and *cis*-acting elements of the *PtFAD* members. In addition, we evaluated the expression patterns of the *PtFAD* genes and UFA contents under abiotic stresses. Those results may lay a critical basis for understanding the biological roles of FADs in response to abiotic stresses and may also contribute to accelerating the process of the genetic improvement of poplar or other species.

## 2. Results

### 2.1. Genome-Wide Identification of FAD Members in P. trichocarpa

A BLASTP analysis of the *P. trichocarpa* genome was executed to investigate the FAD-related proteins. Thirty-two putative *FAD* genes were identified from the *P. trichocarpa* genome, and *Potri.T034100* and *Potri.T180100* were eliminated from the *FAD* candidates due to their localization on the scaffolds (Appendix A). The Pfam, including (PF00487) and (PF03405), were also used to identify the *PtFAD* candidates, and the 23 *FAD* candidates were identified from the *P. trichocarpa* genome, when the incomplete *PtFADs* (*Potri.011G137600* and *Potri.018G107700*) were eliminated (Appendix A). In conclusion, 23 *FAD* genes were identified and renamed as *PtFAD2s*, *PtFAD3/7s/8s*, *PtFAD5s*, *PtFAD6s*, *P. trichocarpa* delta-8-fatty acid desaturase (*PtSLD1/2*), *P. trichocarpa* dihydroceramide desaturase (*PtDSD1/2*), and *PtFAB2s,* based on their similarity with *AtFADs* and *OsFADs* (Appendix A). The *PtFAD* genes ranged from 984 bp (*PtFAD2.1*) to 1359 bp (*PtFAD8.2*) in length, and the corresponding amino acids ranged from 327 aa (PtFAD2.1) to 452 aa (PtFAD8.2) in length. The molecular weights of these PtFADs ranged from 38.12 kDa (PtFAD2.1) to 51.81 kDa (PtFAD8.2), with the isoelectric points varying from 5.62 (PtDSD1) to 9.60 (PtFAD5.2) (Appendix A). Subcellular localization predictions of the PtFADs revealed that they are localized on the chloroplast and ER, and the spatial diversity of these proteins is likely associated with the various functions (Appendix A).

### 2.2. Comparative Alignment and Phylogenetic Analysis

A multiple alignment analysis was performed to characterize the conserved domains, and the most distinguished amino acids in the conserved domains, and to determine the conserved domains of the PtFADs. The result of the multi-alignment revealed that the membrane-bound PtFADs contained three relatively conservative histidine boxes (HXXXH, HXXHH, and HXXHH) (Appendix A). In addition, to determine the evolutionary relationship of the FADs in different species, including *A. thaliana* (*At*), *O. sativa* (*Os*), *P. trichocarpa* (*Pt*), and *T. aestivum* (*Ta*) [42], a phylogenetic tree constructed by the neighbor-joining (NJ) method was used to assess the genetic relationship among the PtFADs, AtFADs, OsFADs, and TaFADs. As shown in Figure 1, all the investigated FADs in this study were classified into two subfamilies: the membrane-bound FADs and soluble FADs (FAB). The membrane-bound FADs were divided into six clades (FAD2, FAD3/7/8, FAD5, FAD6, DSD, and SLD), and the FAB2 subfamily was significantly separated from the membrane-bound FAD subfamily. The phylogenetic tree indicated that the FAB2s and FADs might exist in a common ancestor before the divergence of the monocotyledons and dicotyledons.

The FAD2 clade thought of as ω-6 desaturases included six members from *P. trichocarpa*, two members from *A. thaliana*, four members from *O. sativa*, and fifteen members from *T. aestivum*. The FAD6 was also considered as a kind of ω-6 desaturases, and the FAD6 clade included one member from *A. thaliana*, one member from *O. sativa*, and two members from *P. trichocarpa* and *T. aestivum*, respectively (Figure 1). The ω-3 desaturases included FAD3, FAD7, and FAD8. Here, the PtFAD3 included one member, the PtFAD7 comprised PtFAD7.1/7.2, and the PtFAD8 contained PtFAD8.1/8.2 (Figure 1). The SLD clade catalyzed the desaturation of the sphingolipid at the ∆8 position, comprising three members from *P. trichocarpa*, two members from *A. thaliana*, two members from *O. sativa*, and three members from *T. aestivum* (Figure 1). The sphingolipid ∆4 desaturase DSD comprised two members from *P. trichocarpa*, one member from *A. thaliana*, one member from *O. sativa*, and three members from *T. aestivum* (Figure 1). In addition, the FAD5 (ADS) clade comprised nine members from *A. thaliana* and only two from *P. trichocarpa* (Figure 1). Also, no ADS members were identified from rice and *T. aestivum*, suggesting they may have experienced evolutionary losses in the *O. sativa* and *P. trichocarpa*. In addition, most PtFAB2 members were clustered together with the AtFAB2 members, and the PtFAB2, AtFAB2, OsFAB2, and TaFAB2 members were clustered together, indicating that the FAB2s were not significantly divided into monocotyledonous and dicotyledonous groups and may have independent evolutions in these plant species.

### 2.3. Chromosome Mapping and Duplication Events of P. trichocarpa FAD Genes

The chromosomal location of the 23 poplar *FAD* genes revealed their distributions on the 12 chromosomes, except for Chr03, Chr07, Chr09, Chr12, Chr15, and Chr17, where they were not present (Appendix A). The *PtFAD* genes were mapped unevenly on the *P. trichocarpa* chromosomes. The Chr01, Chr10, and Chr11 had three *FAD* genes, whereas four genes were located on the Chr06. The Ch08 and Chr16 had two *FAD* genes, and Ch02, Ch04, Ch05, Ch13, Ch14, and Ch18 each had one *FAD* gene.

The diversity and expansion of the gene families resulted from tandem and segmental duplication events and we investigated the gene duplication within the *PtFAD* genes. The 16 segmental duplication events were determined among the *PtFAD* family members: *PtFAD2.1*/*PtFAD2.4*, *PtFAD2.1*/*PtFAD2.3*, *PtFAD2.3*/*PtFAD2.4*, *PtFAD3*/*PtFAD7.1*, *PtFAD3*/*PtFAD8.2*, *PtFAD3*/*PtFAD8.1*, *PtFAD8.1*/*PtFAD7.1*, *PtFAD8.1*/*PtFAD7.2*, *PtFAD8.1*/*PtFAD8.2*, *PtFAD7.1*/*PtFAD7.2*, *PtFAD7.1*/*PtFAD8.2*, *PtFAD7.2*/*PtFAD8.2*, *PtFAD6.1*/*PtFAD6.2*, *PtSLD1*/*PtSLD3*, *PtDSD1*/*PtDSD2*, *PtFAB2.2*/*PtFAB2.3* (Figure 2), and *PtFAB2.3*/*PtFAB2.4* as tandem duplication was identified in the syntenic *PtFAD* pairs. Therefore, the segmental duplication events occupied prominent roles in increasing the genetic diversity of the poplar *FAD* family gene. To understand the driving pressure for the *PtFAD* family gene evolution, the Ka/Ks values of the syntenic gene pairs were calculated. The values of Ka/Ks among the *PtFAD* gene pairs were less than one (Appendix A), which indicated that the *PtFAD* members have mainly experienced negative selection. The slow evolutionary process contributes to keeping the conserved functions of the *PtFAD* members.

To further investigate the evolutionary process among the dicotyledonous and monocotyledons, the duplicated gene pairs among the *PtFAD*, *AtFAD*, *OsFAD*, and the *Salix purpurea* FAD (*SpFAD*) members were identified (Figure 3). A total of 14 duplicated gene pairs between *P. trichocarpa* and *A. thaliana* were determined (Appendix A). Also, the following 14 gene pairs between *P. trichocarpa* and *O. sativa* were found (Appendix A). In addition, 28 collinearity gene pairs between *P. trichocarpa* and *S. purpurea* were found, 3 of which belonged to the syntenic *FAB* gene pairs, and the rest were membrane-bound *FAD* gene pairs (Appendix A). Although there is no identification and characterization of the *S. purpurea FAD* family members, the collinearity gene pairs might lay a foundation for studying the *SpFAD* family gene. There were more collinearity gene pairs between *P. trichocarpa* and *S. purpurea*, which might be associated with both poplar and willow, belonging to the Salicaceae. The various *PtFAD* genes have syntenic relationships with three to five willow genes, which suggests that these genes may play essential roles in the evolution of the *FAD* family gene.

### 2.4. Exon–Intron Distribution, Conserved Motif, and 3D Structure Analysis

The *FAD* gene structures were analyzed to gain insight into the exon–intron distributions of the *FADs*. The 47 *FAD* genes were clustered into seven clades, with highly similar patterns of gene structures identified for the *PtFAD* and *AtFAD* genes clustered into the same clade (Figure 4). For the *FAD3*/*7*/*8* cluster, the *FAD3*/*7*/*8* members contained eight exons and seven introns (Appendix A). Also, the *PtFAD3*/*7.2/8.1* has a longer UTR than other *FAD3*/*7*/*8* members. All the *FAD2* members contained one exon except *PtFAD2.1*, which had two exons (Appendix A). In the *FAD6* clade, the *PtFAD6.1* and *6.2* had eight exons and seven introns, while the *AtFAD6* possessed ten exons and nine introns (Appendix A). For the *FAD5* clade, all the *FAD5* contained five exons and four introns, and the *FAD5* members were more dominantly similar in their exon–intron distributions (Appendix A). Specifically, the *AtDSD1* and *PtDSD2* had two exons and an untranslated region (UTR), while the *PtDSD1* contained three exons and was not composed of a UTR. In general, the *SLD* members only contained a single exon and no intron. Most of the FABs harbored three exons in the FAB subfamily, except the *PtFAB2.1* and *AtFAB2.3/2.4*, which had two and four exons, respectively (Appendix A). Similar gene structures of the *FADs* were identified in the same clade, suggesting the exon–intron distribution supports the phylogenetic classification of the *FADs*.

The Multiple Em for Motif Elicitation (MEME) server was used to identify the conserved motifs of the membrane-bound FADs and FAB2s. According to the results, the motifs 1–10 were detected in the FAD family members. Also, the evaluation of motifs 1–10 with the Pfam indicated that motifs 1–7 and motif 10 are related to the *FAD* gene family (Appendix A). The amino acids of ten motifs ranged from 21 to 50, while the sites ranged from 12 to 47 (Appendix A). The FAB2s contain motifs 1, 3, 5, 6, 7, and 10, except for the AtFAB2.4, which harbored the additional motif 2 (Figure 5). The DSD clade had only one motif (motif 3), and the FAD5 and SLD contained two motifs, 2 and 3, except the AtSLD1, which contained motif 4. The clades FAD3/7/8 and FAD2 contained motifs 2, 3, 4, 8, and 9, which indicated that these clades have a close phylogenetic relationship (Figure 5). For the FAD6 clade, motifs 2, 3, and 4 were harbored in the PtFAD6s, while the AtFAD6 was composed of motifs 2, 3, 4, and 8. There is a dominant divergence in the conserved motifs between the membrane-bound FAD and FAB2 (Figure 5). For instance, only motif 3 consensually presented in all the FAD members, while motifs 1, 5, 6, 7, and 10 existed only in the FAB2, suggesting the FAD family members may be involved in two directions in the process of evolution; one is the FAB subfamily, and the other is the membrane-bound FAD subfamily.

The FAD members have been documented to be associated with environmental stresses by introducing a double bond to the formation of UFAs. In order to understand the feathers of the PtFAD protein structures, the 3-dimensional (3D) structures of the PtFADs were generated using the SWISS-MODEL. The 3D structures of the PtFADs and AtFADs showed that the PtFADs and AtFADs are composed of alpha helices, random coils, beta-strands, and the PtFADs and AtFADs clustered into the same clade share similar 3D structures, which provided insight into the function analysis of the PtFAD proteins (Appendix A).

### 2.5. Detection of FAD Cis-Acting Elements

To discover the regulation mechanisms of the FAD member expression patterns, the PlantCARE website was used to investigate the 2 kb upstream promoters of the *PtFAD* and *AtFAD* genes. Various *cis*-acting elements were found, and each *FAD* gene was composed of divergent *cis*-acting elements (Appendix A). Three different *cis*-elements were identified in the *FAD* gene promoters. The first is associated with the plant growth and development, including the light-responsiveness, seed-specific regulation, circadian control, differentiation of the palisade mesophyll cells, and cell cycle regulation. The second is the stress response, including the MeJA-responsiveness, abscisic acid-responsiveness (ABRE), wound-responsiveness, auxin-responsiveness, low-temperature responsiveness, defense and stress-responsiveness, SA-responsiveness, GA-responsiveness, and drought-inducibility. These are related to specific biological processes, including the elements involved in the endosperm expression, zein metabolism regulation, flavonoid biosynthesis regulation, and cell cycle regulation. Most of the *PtFAD* and *AtFAD* promoters harbored the elements involved in the light and hormone responses. For example, the *PtFAD3s* were abundant with the *cis*-acting elements of hormone responsiveness. Also, the *cis*-acting elements involved in flavonoid biosynthesis regulation were identified in five *AtFADs* (*AtADS1*, *AtADS7*, *AtADS5*, *AtADS4*, *AtFAD6)*, and *AtFAB2.2* and one *PtFAD* (*PtDSD2*) promoter, showing that these genes may play essential roles in flavonoid biosynthesis. The promoters of the *AtDSD1*, *PtFAD2.1*, and *PtFAD6.2* contained *cis*-acting elements participating in the seed-specific regulation, suggesting that these genes may have close associations with the molecular regulation of seed development. The wound-responsive element was detected in the *AtFAD7*, *PtDSD2*, and *PtFAD2.4*, indicating that these genes might be mainly involved inwewound response. In summary, various kinds of the *cis*-acting elements were identified in the *PtFAD* and *AtFAD* genes, indicating that the *PtFAD* and *AtFAD* genes may participate in responding to some stimulus.

### 2.6. Interaction Network and Gene Ontology (GO) Annotation of FADs

To better understand the characterizations and profiles of the *FAD* genes, interaction networks and GO annotation were performed to gain more insight into the putative function of the FADs. The results showed that 70 Arabidopsis proteins interacted with 24 AtFADs (Appendix A). For example, as an SA receptor, NPR1 (non-expressor of pathogenesis-related genes 1) plays an important role in regulating plant disease resistance [43]. KASI, beta-ketoacyl-ACP synthase I, significantly participates in C_4_–C_16_ FA biosynthesis and lipid metabolism [44]. CYP38 (Cyclophilin 38) plays an essential role in the assembly and stability of the photosynthetic system II [45]. In addition, 23 PtFADs were predicted to interact with 59 poplar proteins (Appendix A). For example, FAB1, phosphatidylinositol-3-monophosphate5-kinases, participates in vacuole/lysosome homeostasis and transporting various proteins to the vacuole [46]. ACC, acetyl-CoA carboxylase, is involved in ethylene biosynthesis [47]. SAY1, steryl acetyl hydrolase 1, plays a part in lipid metabolism and significantly influences lipid homeostasis [48]. AOX, allene oxide synthase, catalyzes the UFAs to generate important JA precursors. In conclusion, the interaction network analysis of the AtFADs and PtFADs predicted that these proteins were closely associated with the FA and lipid metabolisms and participated in various stress responses.

A GO annotation analysis was performed to improve understanding of the pathway involved in the *FAD* genes. The results indicated that the *AtFADs*, *OsFADs*, and *PtFADs* were dominantly enriched in divergent GO terms, mainly “molecular function” and “biological process” (Appendix A). In the category of “biological process”, the *AtFADs*, *OsFADs*, and *PtFADs* were mainly enriched in the lipid metabolic process (GO:0006629), obsolete oxidation-reduction process (GO:0055114), fatty acid metabolic process (GO:0006631), and monocarboxylic acid metabolic process (GO:0032787). In the category of “molecular function”, the *AtFADs*, *OsFADs*, and *PtFADs* were significantly involved in the oxidoreductase activity (GO:0016717, GO:0016705, and GO:0016491), and acyl–acyl-carrier-protein desaturase activity (GO:0045300). It is suspected that the FADs participate in diverse biological processes, including the FA and lipid metabolisms, and occupy various molecular functions containing the redox and desaturation activities. In addition, the protein interaction relationship contributes to exploring the putative signaling pathway or physiological process of the FADs. Therefore, a GO annotation analysis was performed to analyze the *FAD* and interaction-related protein genes. As shown in Figure 6, the *PtFAD* and interaction-related protein genes were mainly involved in three GO terms: “molecular function”, “cellular component”, and “biological process”. In the category of cellular components, the *PtFAD* and interaction-related protein genes significantly enriched the integral components of the membrane (GO:0016021 and GO:0031224), which suggested that the PtFADs may also be associated with the assembly stability of the membrane.

### 2.7. Expression Profiles of Poplar FAD Genes in Diverse Tissues

The FADs introduce a double bond into the FAs acyl chain leading to the UFAs involved in plant growth and development. To investigate the functional role of the *PtFAD* genes, a real-time quantitative PCR (RT-qPCR) was used to analyze the expression profiles of the diverse tissues, including the mature leaves (ML), young leaves (YL), upper region of the stems (US), lower region of the stems (LS), and roots. The results showed that each *PtFAD* gene has one specific expression pattern.

In *P. trichocarpa*, the relative mRNA levels of the *PtFAD8.1/8.2* and *PtFAB3.5* were lower and absent from the roots. The relative expression of *the PtFAD3/7.1/7.2* was comparatively higher in the YL and lower in the ML. The *PtFAB2.1*/*2.2* was highly expressed in the roots and stems, while lower accumulations were identified in the ML. The *PtFAB2.3* occupied the highest expression levels in the LS and the lowest in the ML, while the highest expression level of the *PtFAB2.5* was found in the YL and the lowest in the UP (Figure 7A). In ‘Nanlin 895’ (*P. deltoides* × *P. euramericana*), the transcript patterns of the *PtFAD7.1/8.1* showed a high similarity, with the highest expression of abundance in the UP and the lowest in the roots. Also, the high expression levels of the *PtFAB2.1*/*2.5* were observed in the roots, while low expression abundances were detected in the stems. The *PtFAD3*/*8.2* genes had a high abundance of expression in the US, while the lower expression levels were found in the ML and roots, respectively.

Interestingly, the *PtFAB2.2*/*2.3* showed lower expression levels in all the tissues tested, whereas the *PtFAB2.4* showed higher expression accumulations in the YL, US, and roots (Figure 7B). In ‘Shanxinyang’ (*P. davidiana* × *P. bolleana* Loucne), similar expression patterns were identified in the *PtFAD7.2*/*8.2*, with relatively higher expression accumulations in the YL and lower in the roots. Also, the *PtFAD3/7/8* and *PtFAB2.2*/*2.3*/*2.5* genes have similar expression profiles and are highly expressed in the YL. In addition, the *PtFAB2.1* displayed a specific expression profile in a more limited root, while the *PtFAB2.1* had a higher expression level in all the tissues tested except the ML (Figure 7C). The expression analysis suggested that the PtFAD members play a key role in the growth and development of the poplar tissues and their functional divergences.

### 2.8. PtFAD Expression Profiles in Response to Multiple Abiotic Stresses

The external environment greatly influences plant growth and development, and the plant can respond to environmental stress by regulating crucial molecular processes. Various stress-related genes help plants to respond to various stresses under adverse environments. Thus, to investigate whether the PtFAD members play a role in abiotic stress and understand the putative regulation mechanisms, a RT-qPCR was applied to analyze their expression patterns under abiotic stress. As shown in Figure 8A, the expression levels of the *PtFADs* were significantly changed under the abscisic acid (ABA) treatment, and their expression patterns were classified into two groups. The expression levels of the *PtFAD8.2* and *PtFAB2.2*/*2.3*/*2.5* showed long-lasting and continuous improvement under the ABA treatment. The peak of the *PtFAD7.1* transcript level was exhibited at 1 h, while the peak of the *PtFAB2.1* expression was identified at 6 h. In addition, the highest expression abundances of the *PtFAD3*/*8.1* were exhibited at 24 h after the ABA treatment. These findings suggested that the *PtFAD8.2* and *PtFAB2.2*/*2.3*/*2.5* play crucial functions in the ABA signal transduction and promote the poplar’s response to environmental stresses.

For the H_2_O_2_ treatment, the *PtFAD3*/*8.1*/*8.2* and *PtFAB2.2* displayed peak expression at the 48 h time point. The *PtFAB2.4* and *PtFAB2.5* had similar trends within the whole time of the H_2_O_2_ treatment, with higher expression levels within the time course. The *PtFAB2.1* and *PtFAB2.3* expression levels were promoted and reached the highest at 12 h and 24 h, respectively. The mRNA level of the *PtFAD7.2* was upregulated at all time points, while the *PtFAD7.1* expression level at 24–48 h was significantly lower than that at 0–12 h (Figure 8B). The response of the *PtFADs* to the NaCl treatment was more dominant than to the ABA treatment. All the tested *PtFAD* expression levels were significantly upregulated during the NaCl treatment, especially the *PtFAD3/7s/8s* and *PtFAB2.3*/*2.5*. The *PtFAB2.4* reached its expression peak at 24 h, and the *PtFAB2.1* and *2.2* were promoted significantly at 6–12 h (Figure 8C). In response to the PEG_6000_ treatment, the upregulated expression of the *PtFAD3/7.1/8s* and *PtFAB2.1*-*2.3,* and the *PtFAB2.5* was observed, and their expression levels after the PEG_6000_ treatment were higher than that before the PEG_6000_ treatment. Also, the *PtFAB2.4* expression was down-regulated at 0–24 h, and its expression level was lower than that of the control (Figure 8D). These results fully reflect that some PtFADs can respond to abiotic stress in poplar, which provides insight into the adaptability promotion of poplar in different environments.

### 2.9. Cluster Network Analysis of PtFADs and PtLACSs

Under the ABA treatment, the cluster dendrogram showed that the *PtFAD* and *PtLACS* expression patterns were divided into two categories (Figure 9A). The *PtFAD3*/*7.2*/*8s*, *PtFAB2.2*/*2.3*/*2.5,* and *PtLACS6*/*7*/*9s* clustered together in the same category, and the *PtFAD2.3*/*7.1*, *PtFAB2.1*/*2.4*, and *PtLACS1-4*/*8.1* clustered together in the other category. Also, a cluster network analysis of the *PtFAD* and *PtLACS* expression levels under the H_2_O_2_ treatment classified them into two categories (Figure 9B). One is the small cluster dendrogram, including the *PtFAD2.3*/*7.1* and *PtLACS3*/*4*, and the other is the big cluster dendrogram, including most of the *PtFAD* and *PtLACS* members. For example, the *PtFAB2.4*/*2.5* and *PtLACS7*/*9.1* clustered together, suggesting they may be involved in the LCFA biosynthesis and desaturation. The cluster dendrogram was also divided into four categories (Figure 9C). The *PtFAD2.3*/*3*/*8.1* and *PtLACS9.1*, *PtFAB2.3-2.5*, *PtFAD7.2*, and *PtLACS7*/*8.1*, and the *PtFAB2.1*/*2.2*, *FAD7.1*/*8.2*, and *PtLACS2*/*6*/*9.2* clustered together, respectively. The result of the cluster network on the *PtFADs* and *PtLACSs* expression under the NaCl treatment indicate that they play essential roles in response to NaCl stress by regulating the FA biosynthesis and desaturation. For the PEG_6000_ treatment, the dendrogram was clustered into two categories, one of which had the *PtLACS1*/*2*/*9.1* and *PtFAD2.3*/*2.4*; the other contained the rest of *the PtLACS* and *PtFAD* members (Figure 9D). For example, the *PtLACS9.1* clustered with the *PtFAD2.2*/*7.1/8.1*, indicating a close relationship in response to the PEG_6000_ treatment. The result of the cluster network showed that expression patterns of some *PtFADs* under the abiotic stress are consistent with those of some *PtLACSs*, indicating that the PtFADs and PtLACSs involved in the LCFA biosynthesis and desaturation are related to the response to abiotic stress.

### 2.10. UFAs Analysis before and after Osmotic Stress

The nonadecanoate (C19:0) was considered as an internal standard, and the quantitative investigation of the single compounds was quantified relative to an internal standard through the automatic integration of the peak areas [49]. Also, the Chromeleon 7.0 was performed to analyze the characteristic peaks of the individual components of the SFA and UFA. In addition, the online tool NIST 17 (https://www.sisweb.com/software/ms/nist.htm (accessed on 8 April 2022) database) was used to sort the data of the SFA and UFA in poplars. The contents of the SFA and UFA were dominantly changed by the drought and salt stresses. The contents of the SFA, including myristate, pentadecanoate, palmitate, margarate, octadecanoate, icosanoate, behenate, and lignocerate, were significantly down-regulated when the poplars were treated by the drought and salt stresses. Also, under the drought and salt treatments, the contents of the UFAs, including the *cis*-9-tetradecenoate, *cis*-10-pentadecenoate, palmitoleate, *cis*-10-heptadecenoate, linoleic acid, *cis*-8,11,14-eicosatrienoate, *cis*-11-eicosenoate, and nervate, were significantly decreased.

In contrast, the contents of the UFAs, including oleate, linolenic acid, and erucate, were dominantly promoted by the drought and salt stresses (Figure 10). The previous studies proved that the oleic acid and ratio of linolenic acid/linoleic acid are closely associated with the osmotic tolerance and are considered as osmotic-regulating substances against osmotic stress [26,39]. The contents of the oleic acid in poplars cultivated under normal conditions were higher than that in poplars cultivated under the drought and salt treatments (Figure 10). Also, the poplars grown under normal conditions produced total linoleic acid levels 55- or 27-fold higher than those treated with the drought or salt treatment (Figure 10). In addition, the linolenic acid contents were significantly increased when the poplars were treated with the drought and salt treatments (Figure 10), and the value of linolenic acid/linoleic acid in poplars treated by the drought or salt treatment was higher than that under the normal condition. The above results suggested that the UFAs participate in osmotic stress, especially the oleic acid, linoleic acid, and linolenic acid. It is speculated that the accumulations of oleic acid, linoleic acid, and linolenic acid improve the membrane lipid fluidity, which can help the poplar respond to abiotic stress.

## 3. Discussion

The UFA compounds and contents are considered the key factor for plant growth and development, and the FADs play an essential role in regulating the UFA biosynthesis [50]. The previous studies on the FADs were constrained to specific plants, and not much information was available on the FADs families in poplars [8,13,42]. Given the vital function of the FADs, systematic studies on poplar FADs were performed. In the present study, a total of 23 *PtFADs* were identified from the *P. trichocarpa* genome based on the Pfams and alignment with the *AtFADs*. There is no significant difference in the number of *FAD* genes among the poplar, Arabidopsis, and rice. However, the number of *PtFADs* was less than in cotton [51], mustard [52], and wheat [42]. A similar number of *FAD* genes in poplar, compared to Arabidopsis and rice, may result from their diploid genome. There are more FADs in cotton, mustard, and wheat than in poplar, Arabidopsis, and rice, possibly because of their ploidy. Therefore, the number of the *FAD* genes did not correspond to the genome size, which indicate that they experienced divergent duplication events during evolution. Gene expansions play an essential role in generating a family gene, and segmental and tandem duplications are commonly associated with the driving forces in family genes [53]. Consistently, the previous studies showed that 22 *FAD* gene pairs are found in rape, all of which are segmental duplication events [54]. Also, 26 and 126 *TaFAD* gene pairs belonging to tandem and segmental duplication were identified from the wheat genome, respectively [42]. In addition, 25 segmental duplicated gene pairs and three tandem duplicated gene pairs were in the *BjFAD* gene family [52]. Here, 17 collinearity gene pairs were determined among the *PtFADs*, containing 16 gene pairs of segmental duplication event and one gene pair of tandem duplication event, which indicate that the segmental duplication event occupies a prominent role in the *PtFAD* family gene expansion.

The phylogenetic analysis indicated that the FAD family was significantly divided into two subfamilies, including the soluble and membrane-bound FADs, similar to the previous studies [55]. Also, the membrane-bound FADs were further separated into the AD3/7/8, FAD2, FAD6, FAD5 (ADS), DSD, and SLD clades [42,55]. The rice did not contain the FAD5 clade, and the PtFAD5 clade only contained the *PtFAD5.1*/*5.2*, which was smaller than the number of the *FAD5* in Arabidopsis. The soluble and membrane-bound FADs were predicted to be localized in diverse positions; the membrane-bound PtFAD members were localized in the ER, and the PtFAB members were chloroplast-localized, like other plants such as sunflower, sesame, and canola [55]. The exon/intron structure variation is related to gene evolution [56]. The *PtFAD* members clustered into the same clade and shared similar gene distributions and motif compositions, similar to other plant species [42,55]. The oil palm *FAD* promoters were identified to have some *cis*-acting elements involved in stress- and light-responsiveness, and their expression was regulated by low temperature and darkness [57]. Also, SA- and JA-responsive elements were identified in the *BnFAD2-C5* promoters. Inconsistently, their expression was upregulated by the SA and JA inducible [38]. Both the *SeFAD2* and *BnFAD* promoters contained the ABRE, and their transcript was induced by the ABA [58,59]. Here, some hormones- and stress-responsive elements were identified in the *PtFAD* promoters, indicating that the *PtFADs* expression may be affected by hormones and stresses. Also, many transcription factor binding sites, such as the MYB binding sites, were found in the *PtFAD* promoters, suggesting that these transcription factors regulate the *PtFAD* expression.

The divergence of the gene expression plays an important role in the family gene, and the *PtFAD* expression patterns provide an opportunity for exploring their profiles and functions. The mRNA levels of the *FAD2/2-1* from *Arachis hypogaea* were accumulated in the seeds, and the expression levels of the *FAD2-2/6* and *SLD1* were elevated in the leaves [41]. The *BjFAD2s* were expressed in all the tissues tested, and the *CaFAD2s* were mainly accumulated in the flowers and seeds [60]. Some *LuFAD2s*, *LuFAD3s*, and *LuFAB2s* from linseed were highly expressed throughout all stages of the seed development, and the *CsFAD* genes were constitutively expressed in the cotyledons and leaves [37,61]. Here, the *PtFADs* were expressed in all the *P. trichocarpa* tissue tested, and the *PtFAD3.1*/*3.3*/*3.4* expression was significantly accumulated in the YL. Also, the higher transcript levels of the *PtFAB2.1*/*2.2* were found in the roots and stems. The previous studies revealed that various abiotic stresses affect poplar development and production, and the FADs are involved in the plant’s adaptation to various abiotic stresses [62]. Several *BnFAD* genes, such as the *BnADS4.4*/*4.8*/*4.9*, *BnSLD8*, and *BnFAD8.1*, potentially regulate the membrane adaptation to cadmium stress. Also, the *BnADS4.9* expression level was remarkably improved, and the *BnFAD7.4* and *BnADS4.8* mRNA levels were decreased after the salt treatment [8]. In addition, the transcript levels of the maize FAD2.1/2.2 and SLD1–3 were upregulated under cold stress, while the *ZmFAB2.1*/*2.3*/*2.12* expression was significantly down-regulated [13]. In this study, the expression patterns of the *PtFADs* were investigated under various abiotic stresses, and the results revealed that the *PtFAD* genes occupy different stress resistance for divergent stresses. For example, the transcript levels of the *PtFAD8.2* and *PtFAB2.2*/*2.3*/*2.5* were continuously induced under the ABA treatment; the *PtFAD3/7s/8s* and *PtFAB2.3*/*2.5* were significantly upregulated within the course of the NaCl treatment; the dominant accumulations of the *PtFAB2.4*/*2.5* expression were identified after the H_2_O_2_ treatment; and the *PtFAD3/7s/8s* and *PtFAB2.1-2.3*/*2.5* expressions were significantly improved when the poplars were treated by the PEG stress. Thus, based on the above evidence, we speculated that the *PtFAD* expression levels have the dominant changes in response to various stresses and play an essential role in the plant’s stress resistance.

With the development of industrial production with wood as the raw material, the requirement for fast-growth and high-quality poplars has come to the fore again in recent years. Abiotic stress, such as drought and salt, has an adverse effect on tree growth and forest productivity [63]. Thus, the improvement of poplar quality contributes to expanding the planting area of poplars. The FAD catalyzes the formation of the UFAs, including the linolenic and linoleic acids, which are associated with the fluidity of the membrane lipids in response to the defense reactions of plants against environmental stresses. The previous studies have documented that the FADs play key roles in plant FA biosynthesis, development, and environmental stress tolerances [62]. The *BnFAD3* were the dominant marker genes for a high oleic acid content, the AtFAD7/8 were associated with the chloroplast trienoic FA contents, and the overexpressing *LuFAD3/7* promoted the α-linolenic acid content in transgenic rice and tobacco [64,65]. In addition, the overexpressing *CrFAB2* in green alga significantly improved the contents of oleic and linoleic acids, and the overexpression of the cytosolic *FAD3* and the plastic *FAD8* remarkably upregulated the ratio of linolenic and linoleic acids, which changed the tolerance to various abiotic stresses [25,66]. In the olive fruit mesocarp, the transcript level of olive *FAD6* was decreased; the oleic/linoleic ratio was promoted at maturation group 1; and the FA content was decreased at the end of the ripening period [67]. In general, the FAD3/7/8 clade members can convert (Δ9,12) linoleic acid to form (Δ9,12,15) linolenic acid [42]. FAB2, the first desaturase in the desaturation process of the FAs, is responsible for desaturating stearoyl-ACP to form oleoyl-ACP and determines the ratio of SFAs and UFAs [68]. In this study, the NaCl and PEG_6000_ treatments were considered as the osmotic stress, and the expression levels of the *PtFAD3/7/8* and *PtFAB2* members significantly increased after the NaCl and PEG_6000_ treatment. These results indicate that the *PtFAD3/7/8* and *PtFAB2* members could respond to osmotic stress in poplars and play an essential role in the adaptability promotion of poplars in an adverse environment. Also, the GC–MS showed that the value of linolenic acid/linoleic acid was significantly promoted. Following the integration of the results of the *PtFAD3/7/8* and *PtFAB2* expression patterns and the SFA and UFA contents, before and after osmotic stress, we proposed that the *PtFAD3/7/8* and *PtFAB2* have a close association with the osmotic stress response involving regulating the oleic acid content and the ratio of linolenic acid to linoleic acid, and in improving the membrane fluidity in response to osmotic stress. Therefore, based on the gene expression and UFA contents results, we speculated that the PtFAD3/7/8 and PtFAB2 could guide the UFA biosynthesis and modulate the SFA and UFA contents, especially the linolenic and linoleic acids, in response to osmotic stress. Considering the *PtFAD3/7/8* and *PtFAB2* expression profiles and the SFA and UFA contents, the putative mechanism of the PtFAD and PtFAB involved in the osmotic stress responses is proposed. When the poplar is under osmotic stress, the transcript levels of the *PtFAD3/7/8* and *PtFAB2* are promoted, and the activities of the PtFAD3/7/8 and PtFAB2 are increased. The increased accumulation of the linolenic acid/linoleic acid value accelerates the membrane fluidity, thus improving the resistance of the poplar to the osmotic stress.

## 4. Materials and Methods

### 4.1. Comparative Analysis of FADs from Arabidopsis, Rice, and Poplar

The genome databases of *A. thaliana*, *O. sativa*, *P. trichocarpa*, and *S. purpurea* were downloaded from Phytozome (http://www.phytozome.net/ (accessed on 8 April 2022)). The previous studies showed that 24 and 18 FAD members are identified from *A. thaliana* and *O. sativa*, respectively [8,42], and the AtFADs and OsFADs were obtained from the Arabidopsis Information Resource (TAIR, https://www.arabidopsis.org/ (accessed on 8 April 2022)) and the Rice Gene Annotation Project (http://rice.plantbiology.msu.edu (accessed on 8 April 2022)), respectively. For a further analysis of the poplar soluble- and membrane-bound FADs, the AtFADs and OsFADs, as the queries, were used to search the poplar proteome using the BLASTP program with the default parameters; the identifiers >40% and an Evalue ≤ 1 × 10^−10^ were used as the criterion. Also, the HMM profile of the FA desaturase (PF00487) and the FA desaturase 2 (PF03405) was used to verify the PtFAD candidates. The candidates were submitted to the NCBI Conserved Domains Database (https://www.ncbi.nlm.nih.gov/Structure/cdd/cdd (accessed on 8 April 2022)) to determine the conserved domain of the PtFAD members. The characterizations and basic information of the PtFADs were identified using the online ExPASy-ProtParam tool (https://web.expasy.org/protparam/ (accessed on 8 April 2022)), and the subcellular localization of the PtFADs was predicted using Cell-PLoc (http://www.csbio.sjtu.edu.cn/bioinf/Cell-PLoc-2/ (accessed on 8 April 2022)).

### 4.2. Multiple Alignment and Construction of a Phylogenetic Tree

Using the ClustalX software, the AtFADs, OsFADs, TaFADs, and PtFADs were conducted to align with the default parameters. The accession numbers of the PtFADs were renamed, based on their similarity with the AtFADs and OsFADs. Using a NJ method with 1000 bootstrap replicates, the MEGA7.0 was performed to construct the phylogenetic tree on the FADs from the poplar, Arabidopsis, and rice.

### 4.3. Comprehensive Analysis of Chromosome Locations and Gene Duplications

The general feature format version 3 (gff3) annotation file in the poplar genome database was used to identify the chromosome locations of the *PtFAD* genes. The chromosomal mapping image of the *PtFAD* genes was visualized by the TBtools software, based on their starting positions on the poplar chromosomes [69]. The previous studies have defined the criteria for gene duplication, including the length of the matched cover >80% and the identity of the matched regions >80% [8,70]. The tandem and segment duplications, considered the distinguished gene duplications, were performed to analyze the *PtFAD* members. To further assess the evolutionary pressure of the *PtFAD* members, the synonymous (Ks) and nonsynonymous (Ka) of the *PtFAD* gene pairs were calculated, using the TBtools with a simple Ka/Ks calculator. In addition, to investigate the evolutionary relationship among the various species, the MCScanX and BLASTP were performed to determine the gene pairs with a collinearity relationship of the FAD members among the poplar, Arabidopsis, rice, and willow.

### 4.4. Investigation of Gene Structures, Motif Distributions, and Protein Structures

The poplar coding sequence and genome file were applied to investigate the splicing phase of the *AtFADs* and *PtFADs*. The Gene Structure Display Server and TBtools software were used to visualize the structure diagrams of the *AtFADs* and *PtFADs*. Also, the MEME was considered for the conserved motifs of the FAD members, and the Pfam and SMART databases were performed to evaluate the function of the conserved motifs. The integrative structures of the AtFADs and PtFADs were constructed by the SWISS-MODEL based on the iterative template-based fragment. The models from the SWISS-MODEL were further refined and visualized by the Chimera software.

### 4.5. Promoter Analysis and Prediction of Protein Interaction and GO Annotation

The 2-kb upstream promoter sequences of the *FADs* were extracted, using the coding sequence and genome files from the poplar and Arabidopsis genome databases. The promoter sequences of the FADs were submitted to PlantCARE to identify *the cis*-regulatory elements. Also, the STRING database was applied to investigate the interaction networks of the AtFAD and PtFAD proteins. To further understand the putative involvement pathways of the *PtFAD* genes, a GO analysis was performed using the TBtools with a GO-basic file.

### 4.6. Plant Cultivation and Various Stress Treatments

All the poplar cultivars used in this study were taken from the Key Laboratory of Landscape Plant Genetics and Breeding at Nantong University. The poplar seedlings were cultivated in a greenhouse at 23 °C and 74% humidity. The poplar leaves with wounds were placed in a medium for regeneration, and then the poplar seedlings produced were transported to the shooting medium for a strong seedling. At the generation of the third or fourth leaves, the poplar shootings were selected and transferred to the rooting medium. The *P. trichocarpa* plants were cultivated in a woody plant medium (WPM) (pH 5.8), supplemented with 0.1 mg/L indole butyric acid (IBA). The ‘Nanlin 895’ (*P. deltoides* × *P. euramericana*) plants were propagated in 1/2 Murashige and Skoog (MS) (pH 5.8), and the ‘Shanxinyang’ *(P. davidiana* × *P. bolleana* Loucne) plants were cultivated in MS medium (pH 5.8) containing 0.3 mg/L IBA. To identify the transcript profiles of the *PtFAD* genes in different tissues, the mature leaves (ML), young leaves (YL), upper region of stems (US), lower region of stems (LS), and roots of 2-month-old poplars grown in the greenhouse were collected. In addition, to examine the expression patterns of the *PtFAD* genes under various stresses, the tissue culture seedlings of the poplar were treated at 0.2 M ABA, 0.2 M NaCl, 2 mM H_2_O_2_, and 10% PEG_6000_, and all the leaf samples were treated by various treatments and collected at 0, 1, 6, 12, 24, and 48 h. All the harvested samples were immediately frozen in liquid nitrogen and stored at −80 °C. Three experimental repetitions were conducted per sample.

### 4.7. Analysis of PtFAD Gene Expression Patterns

The specific primers of the *PtFAD* genes were designed to identify the expression patterns using a RT-qPCR (Appendix A). The total RNA was extracted using the hexadecyl trimethyl ammonium bromide (CTAB) method, as previously described, [71] for all poplar samples. The concentration and quality of the RNA were investigated using a NanoDrop One/OneC spectrophotometer (Thermo Scientific, Waltham, MA, USA). For the reverse-transcription, the PrimeScript™ RT Master Mix (TaKaRa, Kusatsu, Japan) was applied to reverse-transcribe 1 μg of RNA. The gene expression levels were quantified using the ABI 7500 Fast Real-Time PCR System (Applied Biosystem, Waltham, MA, USA) and the UltraSYBR Green I Mixture (CWBIO, Beijing, China). The RT-qPCR amplification reactions were performed in 20 mL volume with the following procedure: pre-denaturation at 95 °C for 5 min, followed by 40 cycles of 95 °C for 10 s, 60 °C for 30 s, and 72 °C for 30 s. The melting curve conditions were 60 °C for 60 s and 95 °C for 15 s. Three experimental repetitions were conducted per selected sample. The 2^−ΔΔCt^ values were estimated to indicate the relative expression levels of the *PtFAD* genes. The TBtools was used to generate heatmaps according to the 2^−ΔΔCt^ values.

### 4.8. Linkage Clustering Analysis of PtFAD and PtLACS Genes

Long-chain acyl-CoA synthetases (LACSs) catalyze FAs to form fatty acyl-CoA thioesters and play a vital role in FA metabolism [72]. Based on the expression patterns of the *PtFADs* and *PtLACSs* under the ABA, NaCl, H_2_O_2_, and PEG_6000_ treatments, the linkage clustering analysis was performed to explore the putative function of the PtFADs and PtLACSs on the lipid and FA metabolisms. The weighted gene cluster network analysis (WGCNA) package in the R programming language was applied to identify the gene cluster networks [73].

### 4.9. Analysis of SFA and UFA Contents

For the drought-stress of soil-grown poplars, the irrigation for poplars cultivated in the soil for three months was suspended, and the leaves were harvested after the drought treatment for two weeks. For the salt stress, the 3-month-old poplars were irrigated with a 200 mM NaCl solution for two weeks, and then the poplar leaves were collected. The experiments were performed under long-day (16-h light/8-h dark) conditions at 23 °C. Three experimental repetitions were applied, each time consisting of six lines. In addition, the collected poplar samples were freeze-dried for 48 h, and the samples were immersed in 2 mL of 10% acetylchloromethanol and 1 mL of n-hexane, and a total of 3 mL of reaction solvent was incubated for 2 h at 95 °C. Then, a total of 6 mL of 6% potassium carbonate solution was added to the above mixtures, and the n-hexane phase was obtained after centrifugation for 15 min at 4 °C, then, the n-hexane was removed from the n-hexane phase by vacuum concentration. The collected solvents containing the n-hexane and nonadecanoic acid were injected into a gas chromatography–mass spectrometry (GC–MS) (Thermo ISQ7000, Thermo Fisher, Waltham, MA, USA) or GC–flame ionization detector (Thermo Trace1300, Thermo Fisher, Waltham, MA, USA) with a chromatographic column DB-5 (60 m × 0.25 mm × 0.25 µm). In addition, GC–MS was applied to investigate the UFA contents. The GC–MS was performed as follows: (1) The column temperature was kept at 140 °C for 5 min, increased to 180 °C at the rate of 10 °C/min, and then rose to 210 °C at the rate of 4 °C/min. (2) The temperature was maintained at 310 °C for 30 min at the rate of 10 °C/min, and the capillary GC with a flame ionization detector was used to identify the individual components of the fatty acids in poplars. All data were presented as the mean ± standard deviation. A one-way analysis of variance (ANOVA), followed by the Tukey test, was used to determine the significant difference by the SPSS software (SPSS Inc., Chicago, IL, USA).

## 5. Conclusions

In this study, 23 *PtFAD* genes were identified from the *P. trichocarpa* genome. The phylogenetic tree indicated that the PtFADs could be divided into two subfamilies, including the soluble and membrane-bound FADs. The exon–intron compositions and conserved motifs of the PtFADs were similar within the same clade. A total of 17 syntenic gene pairs of the *PtFAD* members were found in the poplar genome, which might have a close association with the *PtFAD* duplication during evolution. Protein interaction analysis revealed that the PtFADs play a crucial role in the UFAs biosynthesis, and the GO analysis showed that the PtFADs are enriched in the molecular function and biological processes. The PtFAD3/7/8 mediated the regulation of the linolenic and linoleic acid levels and were associated with poplar’s resistance to drought and salt stresses. The study on the characterizations and profiles of the *PtFAD* genes provides insight into the potential functions of the *PtFADs* and sheds light on more candidate genes for genetic breeding.

## Figures and Tables

**Figure 1 ijms-23-11109-f001:**
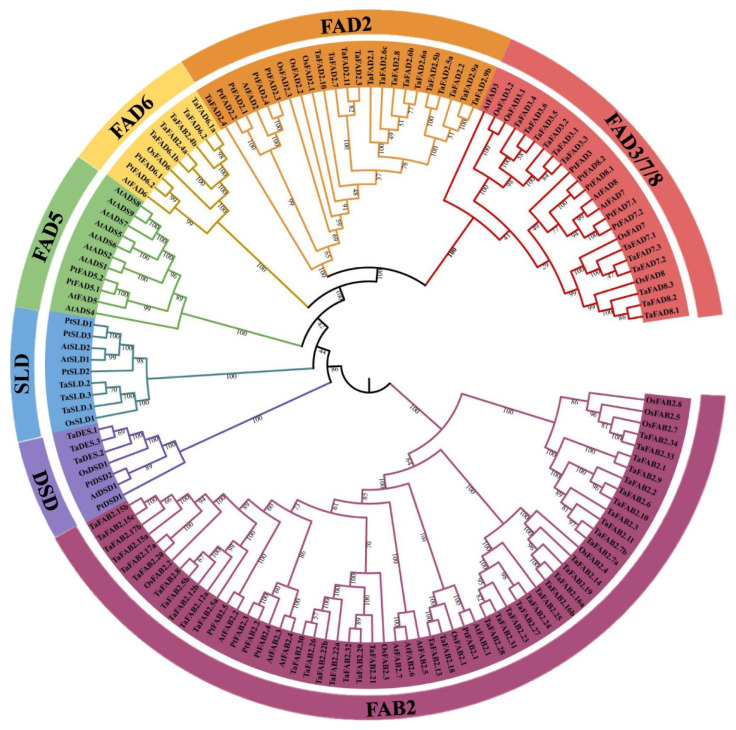
Phylogenetic relationships of the FAD proteins from *Arabidopsis thaliana* (*At*), *Oryza sativa* (*Os*), *Populus trichocarpa* (*Pt*), and *Triticum aestivum* (*Ta*). The PtFAD, AtFAD, and OsFAD proteins were aligned using the ClustalW, and the phylogenetic tree was constructed using the MEGA7.0. The FAD members were divided into seven clades: FAD3/7/8, FAD2, FAD6, DSD, FAD5, SLD, and FAB2, and different clades were indicated by different colors.

**Figure 2 ijms-23-11109-f002:**
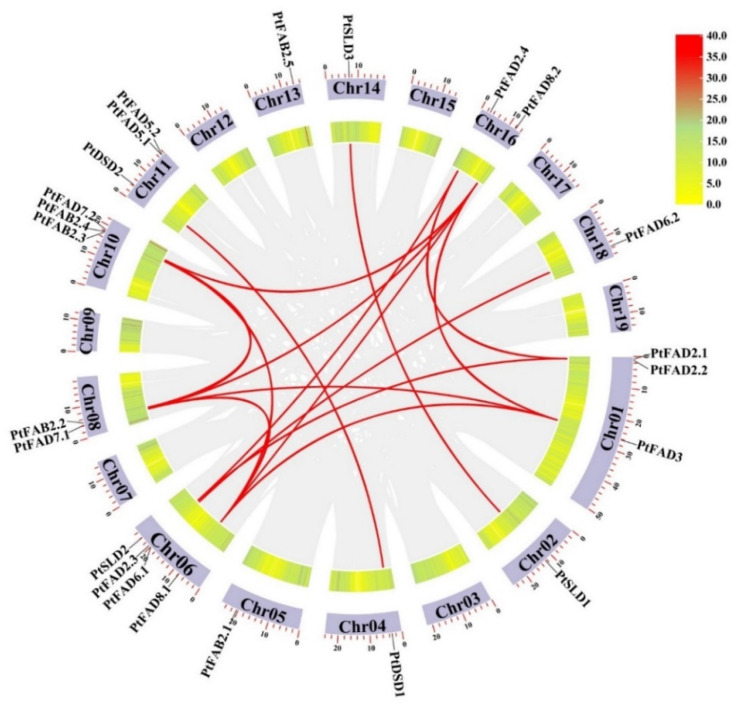
Collinearity analysis of *PtFAD* gene family members. A rectangle represented chromosomes. The chromosomal positions of *PtFADs* were plotted using the TBtools. Individual *PtFAD* gene positions were labeled using the *PtFAD* name, and individual chromosomes were labeled with respective *PtFADs* in the circle. Each dark and red colored curve indicated the gene duplication event of all collinearity gene pairs and *PtFADs* across the chromosomes, respectively.

**Figure 3 ijms-23-11109-f003:**
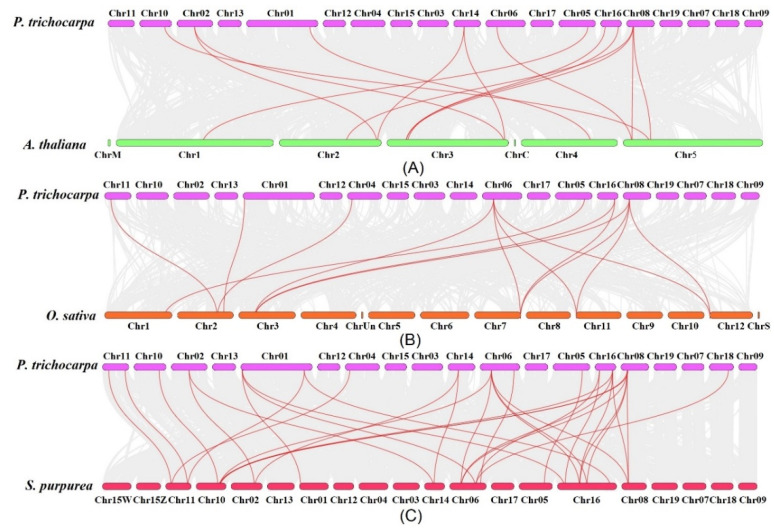
Syntenic relationship of *FAD* genes between *P. trichocarpa* and three other plant species, including *A. thaliana* (**A**), *O. sativa* (**B**), and *Salix purpurea* (**C**). The collinearity gene pairs and *FAD* gene pairs were represented in dark and red lines. The chromosomes of *P. trichocarpa*, *A. thaliana*, *O. sativa*, and *S. purpurea* were remarked by differently colored boxes.

**Figure 4 ijms-23-11109-f004:**
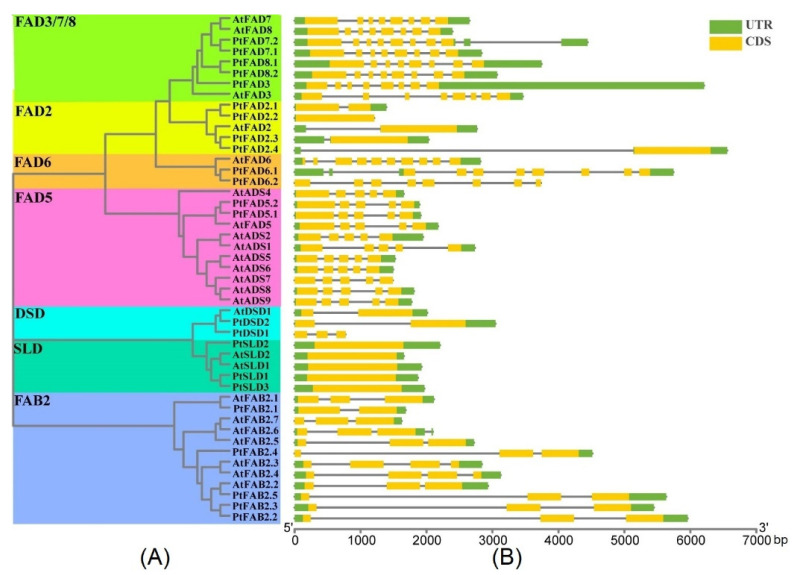
Exon–intron distributions of PtFAD and AtFAD members. (**A**) The unrooted phylogenetic tree was shown, constructed using MEGA7.0 set to the neighbor-joining (NJ) method, and different clades were marked with different color backgrounds. (**B**) Exons, introns, and untranslated regions (UTR) were indicated by yellow rectangles, grey lines, and green rectangles, respectively. PtFAD and AtFAD members were clustered based on a phylogenetic tree. Base pair: bp.

**Figure 5 ijms-23-11109-f005:**
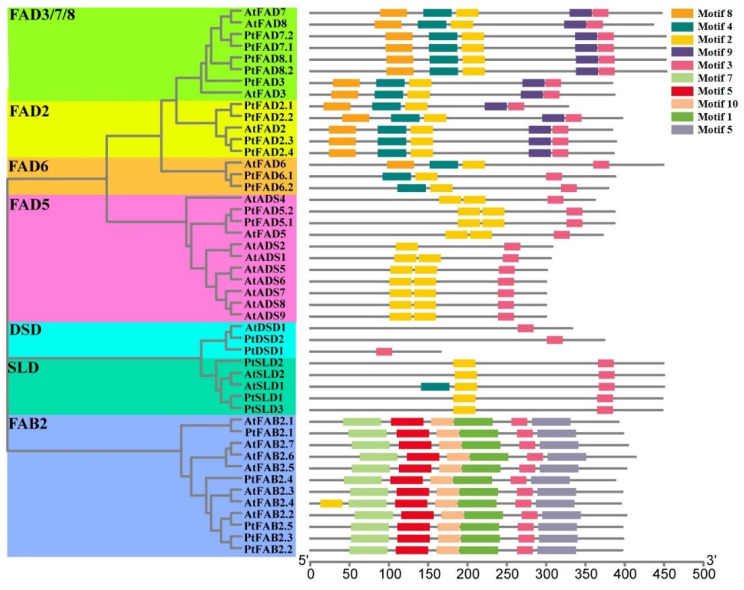
The conserved motifs of PtFAD and AtFAD members. The distribution of conserved motifs in the PtFAD and AtFAD proteins. The unrooted phylogenetic tree was shown, constructed using MEGA7.0 set to the neighbor-joining (NJ) method, and different clades were marked with different color backgrounds.

**Figure 6 ijms-23-11109-f006:**
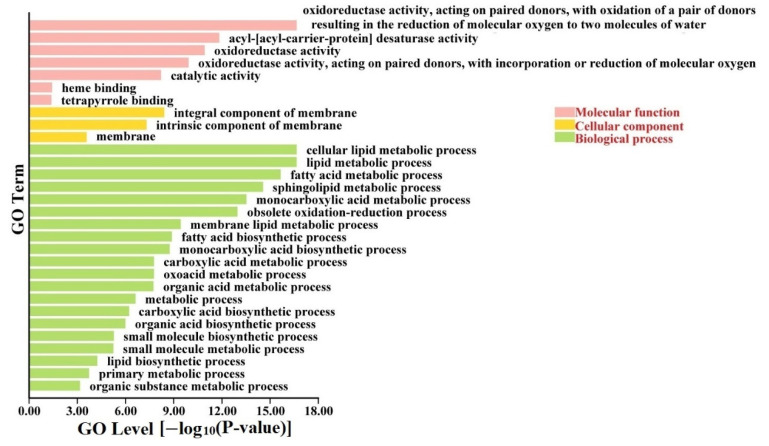
Gene ontology (GO) analysis of *PtFAD* and interaction protein genes. GO analysis was performed using TBtools with a go-basic file, and *PtFAD* and interaction protein genes were enriched in molecular function, cellular component, and biological process.

**Figure 7 ijms-23-11109-f007:**
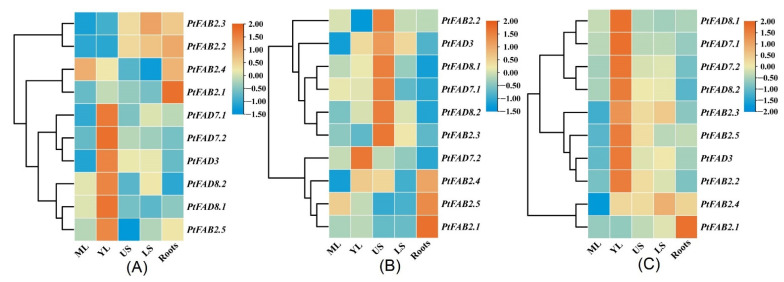
The expression pattern of *PtFAD* genes in various tissues and organs. The RT-qPCR was applied to investigate *PtFAD* expression patterns in *P. trichocarpa* (**A**), ‘Nanlin 895’ (*Populus* × *euramericana* cv.) (**B**), and ‘Shanxinyang’ (*P. davidiana* × *P. bolleana*) (**C**) tissues. The heatmap was generated using TBtools software. The color scale indicated the normalized data, where dark orange represented a high expression level, blue represented a low expression level, and orange represented a medium level. Mature leaves: ML; young leaves: YL; the upper region of stems: US; the lower region of stems: LS.

**Figure 8 ijms-23-11109-f008:**
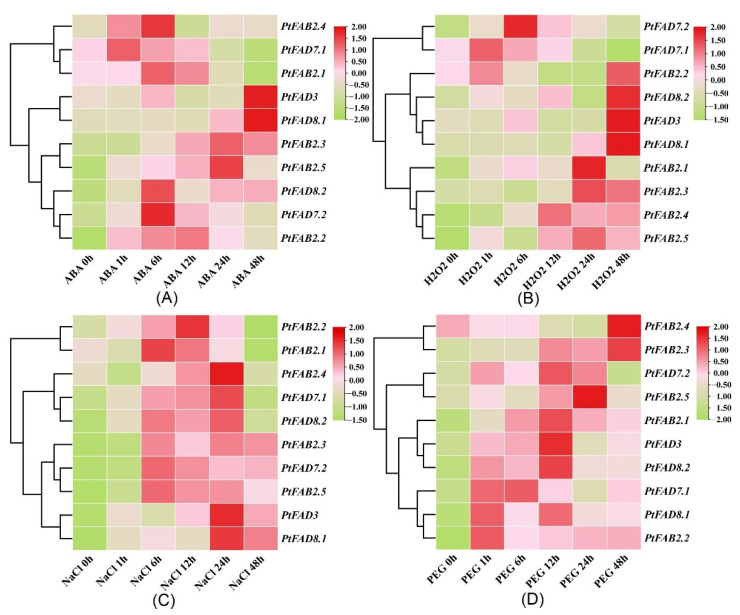
The expression patterns of *PtFAD* genes under ABA (**A**), H_2_O_2_ (**B**), NaCl (**C**), and PEG_6000_ (**D**). The heatmap was generated using TBtools software. The color scale indicated the normalized data, where pink represented a high expression level, light pink represented a low expression level, and light green represented a medium level.

**Figure 9 ijms-23-11109-f009:**
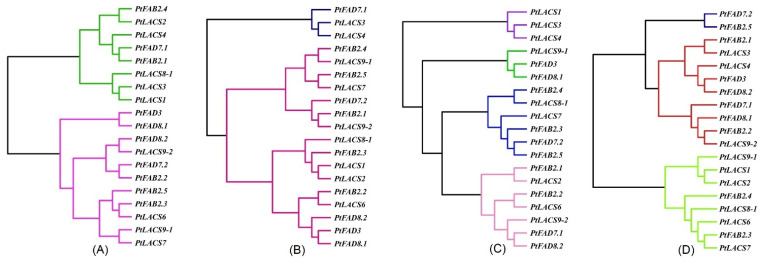
The cluster dendrogram of *PtFAD* and *PtLACS* expression patterns under ABA (**A**), H_2_O_2_ (**B**), NaCl (**C**), and PEG_6000_ (**D**). The data were normalized to *PtActin* (XM-006370951) expression level. Each *PtFAD* and *PtLACS* expression level was calculated based on the corresponding gene mRNA level in poplar without treatments.

**Figure 10 ijms-23-11109-f010:**
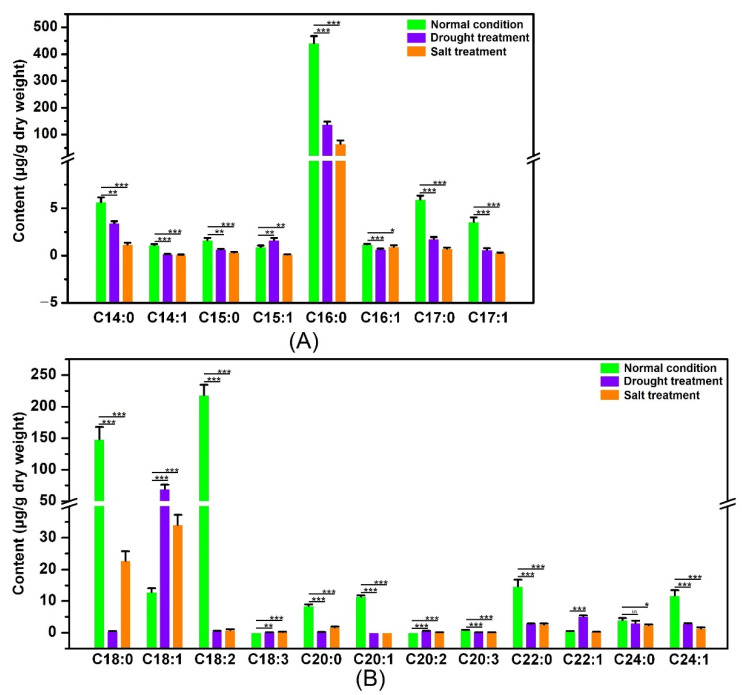
The gas chromatography-mass spectrometry (GC-MS) analysis of the SFA and UFA contents of poplar leaves before and after drought or salt stress. (**A**) The contents of C14-C17 SFAs and UFAs in poplar leaves cultivated under normal conditions and drought or salt stress. (**B**) The contents of C18-C24 SFAs and UFAs in poplar leaves cultivated under normal conditions and drought or salt stress. Three biological replicates were performed per group. * *p* < 0.05, ** *p* < 0.01, *** *p* < 0.001 compared to control poplar. The kinds of SFAs and UFAs included myristate (C_14:0_), *cis*-9-tetradecenoate (C_14:1_), pentadecanoate (C_15:0_), *cis*-10-pentadecenoate (C_15:1_), palmitate (C_16:0_), palmitoleate (C_16:1_), margarate (C_17:0_), *cis*-10-heptadecenoate (C_17:1_), octadecanoate (C_18:0_), oleate (C_18:1_), linoleic acid (C_18:2_), linolenic acid (C_18:3_), icosanoate (C_20:0_), eicosenoate (*cis*-11) (C_20:1_), eicosadienoate (*cis*-11, 14) (C_20:2_), eicosatrienoate (*cis*-8,11,14) (C_20:3_), behenate (C_22:0_), erucate (C_22:1_), lignocerate (C_24:0_), and nervate (C_24:1_).

## Data Availability

Not applicable.

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
