# Peer review of "Genome-Wide Characterization and Expression Analysis of Fatty acid Desaturase Gene Family in Poplar"

_ijms, 2022, doi:10.3390/ijms231911109_

Round 1

Reviewer 1 Report

In this manuscript, the authors performed a detailed bioinformatics analysis of FAD members in poplar, including phylogenetic analysis, chromosome mapping and duplication events analysis, exon-intron distribution, conserved motif, and cis-acting elements analysis, these are classic methods in the gene family study. Authors also analyzed the gene expression pattern of FAD genes and UFA contents under different stress condition. In general, the data presented in this manuscript looks good. But the paper is not presented well with many flaws need clarify and improvements based on the overall assessment. Firstly, English need be improved in the whole manuscript, there are many unclear statements make the manuscript difficult to understand. I suggest sending the paper to a native English speaker for language revision. There are also some other problems as follows:

1.       The title of the manuscript is not perfectly fit the context of the paper. In this paper, more than half of the context are related to genome-wide characterization of this gene family in poplar, I suggest “Genome-wide characterization and expression analysis of fatty acid desaturase gene family in poplar” this kind of title is more appropriate for this paper.

2.       There are many places in the context which have grammatical mistakes, for example: line 42 “Their biosynthesis and regulation play an important in the basic metabolic….”  Should be “play an important role in….”. Line 48 The UFAs considered the essential constituents for bio-membranes to improve….” Should be The UFAs are considered to be the essential…..”. line 51, “Two categories of FADs have been identified 51 in plants; one is soluble FAD broadly distributed on plant cell plastids. Another is mem-52 brane-bound FADs mainly….” Should be one is… the other one is….”. Line 62 The glycolytic pathway can provide the acetyl-coenzyme A usually in the description of a established fact, it is not necessary use “can”, “The glycolytic pathway provides the acetyl-coenzyme A….” is more appropriate. Line 87-88 The FAD2 and FAD3 were exposed on the cytosolic side of ER membranes and catalyzed the conversion of C18:1 PC-bound oleate acid to C18:2 linoleic acid and C18:3 linolenic acid, respectively. Through the whole context, there are many places the authors use the English tense incorrectly. Here should use the present tense, but the authors use the past tense.

3.       Line95 -103, In this manuscript, there is no study results related to wounding, but here and some other places the authors gave a long statements and discussion related to wounding.  I suggest delete some of this background.

4.       Line 136-137 “Those results may lay a critical basis for understanding the biological 136 roles of FAD in response to abiotic stresses and may also contribute to accelerating the process of genetic improvement” this statement is not complete, should be “……. accelerating the process of genetic improvement of poplar or other species”.

5.       In Figure 1, as wheat has A,B and D 3 sub-genome, I suggest for every wheat FAD gene, for example TaFAD8.1, TaFAD8.2, TaFAD8.3, distinguish them with A,B,D, like TaFAD8A, TaFAD8B and TaFAD8D.

6.       Line 194-195, “Also, no ADS members were identified in rice and wheat, suggesting they may experience an evolutionary loss in rice and poplar” check this sentence and correct it.

7.       Line 208, should be 23 poplar FAD genes

8.       Line 248, gene structures identified for PtFAD, AtFAD, and OsFAD genes clustered into the same 248 clade (Figure 4). Make sure that if OsFAD genes are really included in the analysis?

9.       Line 274, “The FAB2s contained motifs…. “Should be contains, check other places and revise them

10.   Line 322 In summary, the PtFAD and AtFAD genes seem to occupy their cis-acting elements, indicating….” Not clear this sentence.

11.   In figure 7, line 373 oranges should be organs

12.   Line 412, “The transcript levels of PtFAD7.1 and PtFAB2.1 were improved first, followed by a 412 gradual down-regulation” this sentence should be improved.

13.   Line 435 “H2O2” is not correct.

14.   The discussion is not well written, should be improved, more information related to the function of FAD genes which was studied in poplar by previous papers should be discussed.

Line 678-679, more details should be provided for the samples collected for transcript analysis. Like how old is the plants used to collect sample. How to define ML and YL.

Author Response

In this manuscript, the authors performed a detailed bioinformatics analysis of FAD members in poplar, including phylogenetic analysis, chromosome mapping and duplication events analysis, exon-intron distribution, conserved motif, and cis-acting elements analysis, these are classic methods in the gene family study. Authors also analyzed the gene expression pattern of FAD genes and UFA contents under different stress condition. In general, the data presented in this manuscript looks good. But the paper is not presented well with many flaws need clarify and improvements based on the overall assessment. Firstly, English need be improved in the whole manuscript, there are many unclear statements make the manuscript difficult to understand.

Response: Thank you for the reviewers’ comments concerning our manuscript. Those comments are all valuable and very helpful for revising and improving our paper, as well as the important guiding significance to our research. The authors have studied comments carefully and made corrections, which we hope meet with approval.

I suggest sending the paper to a native English speaker for language revision. There are also some other problems as follows:

  1. The title of the manuscript is not perfectly fit the context of the paper. In this paper, more than half of the context are related to genome-wide characterization of this gene family in poplar, I suggest “Genome-wide characterization and expression analysis of fatty acid desaturase gene family in poplar” this kind of title is more appropriate for this paper.

Response: Thanks for your valuable comments. The authors have revised accordingly.

  1. There are many places in the context which have grammatical mistakes, for example: line 42 “Their biosynthesis and regulation play an important in the basic metabolic….”  Should be “play an important role in….”. Line 48 The UFAs considered the essential constituents for bio-membranes to improve….” Should be “The UFAs are considered to be the essential…..”. line 51, “Two categories of FADs have been identified 51 in plants; one is soluble FAD broadly distributed on plant cell plastids. Another is mem-52 brane-bound FADs mainly….” Should be “one is… the other one is….”. Line 62 “The glycolytic pathway can provide the acetyl-coenzyme A” usually in the description of a established fact, it is not necessary use “can”, “The glycolytic pathway provides the acetyl-coenzyme A….” is more appropriate. Line 87-88 The FAD2 and FAD3 were exposed on the cytosolic side of ER membranes and catalyzed the conversion of C18:1 PC-bound oleate acid to C18:2 linoleic acid and C18:3 linolenic acid, respectively. Through the whole context, there are many places the authors use the English tense incorrectly. Here should use the present tense, but the authors use the past tense.

Response: Thanks for your valuable comments. The authors have revised accordingly.

  1. Line95 -103, In this manuscript, there is no study results related to wounding, but here and some other places the authors gave a long statements and discussion related to wounding.  I suggest delete some of this background.

Response: Thanks for your valuable comments. The authors have deleted some of this background accordingly.

  1. Line 136-137 “Those results may lay a critical basis for understanding the biological 136 roles of FAD in response to abiotic stresses and may also contribute to accelerating the process of genetic improvement” this statement is not complete, should be “……. accelerating the process of genetic improvement of poplar or other species”.

Response: Thanks for your valuable comments. The authors have revised accordingly.

  1. In Figure 1, as wheat has A,B and D 3 sub-genome, I suggest for every wheat FAD gene, for example TaFAD8.1, TaFAD8.2, TaFAD8.3, distinguish them with A,B,D, like TaFAD8A, TaFAD8B and TaFAD8D.

Respond: Thanks for your valuable comments. As the reviewer pointed out, the wheat has A,B and D 3 sub-genome. The Hajiahmadi et al (2020) has Identified and renamed the fatty acid desaturase genes in wheat. Therefore, the authors adopt the naming method of this article.

Reference:

Hajiahmadi, Z.; Abedi, A.; Wei, H.; Sun, W.; Ruan, H.; Zhuge, Q.; Movahedi, A. Identification, evolution, expression, and docking studies of fatty acid desaturase genes in wheat (Triticum aestivum L.). BMC Genomics. 2020, 21, 1-20.

  1. Line 194-195, “Also, no ADS members were identified in rice and wheat, suggesting they may experience an evolutionary loss in rice and poplar” check this sentence and correct it.

Response: Thanks for your valuable comments. The authors have revised accordingly.

  1. Line 208, should be 23 poplar FAD genes

Response: Thanks for your valuable comments. The authors have revised accordingly.

  1. Line 248, gene structures identified for PtFAD, AtFAD, and OsFAD genes clustered into the same 248 clade (Figure 4). Make sure that if OsFAD genes are really included in the analysis?

Response: Thanks for your valuable comments. The authors have revised accordingly.

  1. Line 274, “The FAB2s contained motifs…. “Should be contains, check other places and revise them

Response: Thanks for your valuable comments. The authors have revised accordingly.

  1. Line 322 “In summary, the PtFAD and AtFAD genes seem to occupy their cis-acting elements, indicating….” Not clear this sentence.

Response: Thanks for your valuable comments. The authors have revised accordingly.

  1. In figure 7, line 373 oranges should be organs

Response: Thanks for your valuable comments. The authors have revised accordingly.

  1. Line 412, “The transcript levels of PtFAD7.1 and PtFAB2.1 were improved first, followed by a 412 gradual down-regulation” this sentence should be improved.

Response: Thanks for your valuable comments. The authors have revised accordingly.

  1. Line 435 “H2O2” is not correct.

Response: Thanks for your valuable comments. The authors have revised accordingly.

  1. The discussion is not well written, should be improved, more information related to the function of FAD genes which was studied in poplar by previous papers should be discussed.

Response: Thanks for your valuable comments. The authors rewrote the discussion.

Line 678-679, more details should be provided for the samples collected for transcript analysis. Like how old is the plants used to collect sample. How to define ML and YL.

Response: Thanks for your valuable comments. The authors have revised accordingly.

Reviewer 2 Report

About the paper, with my knowledge in relation to functional genomics, I think this is a complete article because includes both genome-wide and expression patterns analysis of FAD genes. I think the quantity of study work is enough to analyze the function of FAD genes in Poplar against abiotic stresses through phylogenetic, chromosome distribution, collinearity relationship, selection pressure, gene structure, cis-acting elements, and the expression patterns of FADs and UFA contents under drought or salt stress.

Line617. Why select A. thaliana, O. sativa, and S. purpurea to promote the comparative analysis? Line616. The subtitle, comparative analysis of FADs from Arabidopsis, rice, and polar. No S. purpurea? Subtitle 4.1, where does the sequence of FADs come from, and how many FADs were selected to further analysis? I think this information should be written clearly in the materials and methods.

Line641. Cite the reference.

Line678. What ML, YL, US, LS?

Line687. qRT-PCR or RT-qPCR?

Subtitle 4.8. How linkage clustering analysis of PtFAD and PtLACS genes? It is not clear in this part.

Line143. Poplar genome replaced by P. trichocarpa genome?

The second paragraph in sub-title 2.2 should be shortened and condensed statement.

Line216. Please confirm the number of syntenic gene pairs in figure 2.

Line242. In the first appearance, Willow was replaced by Willow (Latin name). other similar places also would be revised.

Figure 5 is not clear.

Sub title2.9, cluster network analysis of PtFADs and PtLACS is one of the important parts of the paper. The figure of cluster network analysis would be cite into the main text body, such as supplementary figure S8. But I could not find the figure S8, I uncertainly it is suitable cited in the main text body.

In the introduction, I suggest to supply some contents about the role of UFAs under abiotic stresses.

Sub title2.9 and 2.10. The contents with cited references would be replaced in discussion?

Line611, up-regulation of linolenic acid/linoleic acid value would be replaced by increased-accumulation of linolenic acid/linoleic acid value?

Figure 9. ANOVA and multiple comparison were used to determine the significant differences of fatty acid after drought or salt stress? The method should be explain in the part of materials and methods.

The correlation between expression patterns of FADs and the contents of fatty acid would be a more in-depth discussion.

Author Response

About the paper, with my knowledge in relation to functional genomics, I think this is a complete article because includes both genome-wide and expression patterns analysis of FAD genes. I think the quantity of study work is enough to analyze the function of FAD genes in Poplar against abiotic stresses through phylogenetic, chromosome distribution, collinearity relationship, selection pressure, gene structure, cis-acting elements, and the expression patterns of FADs and UFA contents under drought or salt stress.

Response: Thank you for the reviewers’ comments concerning our manuscript. Those comments are all valuable and very helpful for revising and improving our paper, as well as the important guiding significance to our research. The authors have studied comments carefully and made corrections, which we hope meet with approval.

Line617. Why select A. thalianaO. sativa, and S. purpurea to promote the comparative analysis? Line616. The subtitle, comparative analysis of FADs from Arabidopsis, rice, and polar. No S. purpurea? Subtitle 4.1, where does the sequence of FADs come from, and how many FADs were selected to further analysis? I think this information should be written clearly in the materials and methods.

Response: Thanks for your valuable comments. The purpose of the manuscript was to evaluate the characterizations and profiles of PtFADs. According to homologous alignment with Arabidopsis and rice FADs, the putative PtFAD candidates were identified according to homologous alignment. The AtFAD and OsFAD family members have been identified (Xu et al., 2019; Hajiahmadi et al., 2020), but no information on willow GAPDH family members. In addition, the poplar, Arabidopsis, and willow are dicotyledon, whereas the rice is a monocotyledon. To investigate the evolutionary process among the dicotyledonous and monocotyledons, we identified the collinearity relationship of FADs among P. trichocarpa, A. thalianaO. sativa, and S. purpurea.

Reference:

Xu, L.; Zeng, W.; Li, J.; Liu, H.; Yan, G.; Si, P.; ... Zhou, W. Characteristics of membrane-bound fatty acid desaturase (FAD) genes in Brassica napus L. and their expressions under different cadmium and salinity stresses. Environ Exp Bot. 2019, 162, 144-156.

Hajiahmadi, Z.; Abedi, A.; Wei, H.; Sun, W.; Ruan, H.; Zhuge, Q.; Movahedi, A. Identification, evolution, expression, and docking studies of fatty acid desaturase genes in wheat (Triticum aestivum L.). BMC Genomics. 2020, 21, 1-20.

Line641. Cite the reference.

Response: Thanks for your valuable comments. The authors have revised accordingly.

Line678. What ML, YL, US, LS?

Response: Thanks for your valuable comments. The authors have revised accordingly.

Line687. qRT-PCR or RT-qPCR?

Response: Thanks for your valuable comments. The authors have revised accordingly.

Subtitle 4.8. How linkage clustering analysis of PtFAD and PtLACS genes? It is not clear in this part.

Response: Thanks for your valuable comments. The authors have revised accordingly.

Line143. Poplar genome replaced by P. trichocarpa genome?

Response: Thanks for your valuable comments. The authors have revised accordingly.

The second paragraph in sub-title 2.2 should be shortened and condensed statement.

Response: Thanks for your valuable comments. The authors have revised accordingly.

Line216. Please confirm the number of syntenic gene pairs in figure 2.

Response: Thanks for your valuable comments. The authors have revised accordingly.

Line242. In the first appearance, Willow was replaced by Willow (Latin name). other similar places also would be revised.

Response: Thanks for your valuable comments. The authors have revised accordingly.

Figure 5 is not clear.

Response: Thanks for your valuable comments. The authors have revised Figure 5.

Sub title2.9, cluster network analysis of PtFADs and PtLACS is one of the important parts of the paper. The figure of cluster network analysis would be cite into the main text body, such as supplementary figure S8. But I could not find the figure S8, I uncertainly it is suitable cited in the main text body.

Response: Thanks for your valuable comments. The authors have added Figure 9 on cluster network analysis of PtFADs and PtLACSs.

In the introduction, I suggest to supply some contents about the role of UFAs under abiotic stresses.

Response: Thanks for your valuable comments. The authors have supplied some content about the role of UFAs under abiotic stresses in the introduction.

Sub title2.9 and 2.10. The contents with cited references would be replaced in discussion?

Response: Thanks for your valuable comments. The authors have revised accordingly.

Line611, up-regulation of linolenic acid/linoleic acid value would be replaced by increased-accumulation of linolenic acid/linoleic acid value?

Response: Thanks for your valuable comments. The authors have revised accordingly.

Figure 9. ANOVA and multiple comparison were used to determine the significant differences of fatty acid after drought or salt stress? The method should be explain in the part of materials and methods.

Response: Thanks for your valuable comments. The authors have added the ANOVA and multiple comparisons in the part of materials and methods accordingly.

The correlation between expression patterns of FADs and the contents of fatty acid would be a more in-depth discussion.

Response: Thanks for your valuable comments. The authors have revised accordingly.